# Fatigue Analysis of NiTi Rotary Endodontic Files through Finite Element Simulation: Effect of Root Canal Geometry on Fatigue Life

**DOI:** 10.3390/jcm10235692

**Published:** 2021-12-03

**Authors:** Victor Roda-Casanova, Antonio Pérez-González, Álvaro Zubizarreta-Macho, Vicente Faus-Matoses

**Affiliations:** 1Department of Mechanical Engineering and Construction, Universitat Jaume I, 12071 Castelló de la Plana, Spain; vroda@uji.es (V.R.-C.); aperez@uji.es (A.P.-G.); 2Department of Dentistry, Alfonso X el Sabio University, 28691 Madrid, Spain; 3Department of Orthodontics, University of Salamanca, 37008 Salamanca, Spain; 4Department of Stomatology, Faculty of Medicine and Dentistry, University of Valencia, 46010 Valencia, Spain; vicente.faus@uv.es

**Keywords:** endodontic rotary files, finite element analysis, fatigue analysis

## Abstract

This article describes a numerical procedure for estimating the fatigue life of NiTi endodontic rotary files. An enhanced finite element model reproducing the interaction of the endodontic file rotating inside the root canal was developed, which includes important phenomena that allowed increasing the degree of realism of the simulation. A method based on the critical plane approach was proposed for extracting significant strain results from finite element analysis, which were used in combination with the Coffin–Manson relation to predict the fatigue life of the NiTi rotary files. The proposed procedure is illustrated with several numerical examples in which different combinations of endodontic rotary files and root canal geometries were investigated. By using these analyses, the effect of the radius of curvature and the angle of curvature of the root canal on the fatigue life of the rotary files was analysed. The results confirm the significant influence of the root canal geometry on the fatigue life of the NiTi rotary files and reveal the higher importance of the radius of curvature with respect to the angle of curvature of the root canal.

## 1. Introduction

The use of nickel-titanium (NiTi) rotary files for shaping root canals has spread in endodontics during the last decades, in detriment of manual preparation with traditional stainless-steel instruments. The superelasticity of NiTi and its lower Young’s modulus reduce the risk of canal transportation and ledging in the treatment of curved root canals [1]. The superelasticity of the NiTi refers to the capacity of the material for undergoing large elastic deformations that can be restored after the forces producing the deformation are released. During these large deformations of the superelastic material, a phase transformation is induced within the material from austenite to martensite at a nearly constant stress. Due to this superelastic behaviour, files made of NiTi can adapt easily to strongly curved root canals. Successive modifications introduced during the last two decades in these instruments have allowed improving the quality of the cleaning and shaping, as well as saving time for both clinicians and patients [2,3,4]. However, the main problem that persists is the fracture of the files inside the root canal [5].

Fracture of rotary instruments occurs mainly by two different mechanisms, usually referred to as torsion overload and flexural fatigue [6,7]. A torsion overload mechanism corresponds to a static failure and occurs when a section of the file is locked within the canal, and the shank continues to rotate. In this static failure, the file fails because the stress value reaches the elastic limit of the material, and the file undergoes permanent deformations and finally it fractures. Flexural fatigue is a failure mechanism produced mainly by the alternating compressive and tensile stresses and strains that appear in any point of a file rotating inside a curved root canal. This fatigue failure results in a sudden fracture of the file after a certain number of rotations, even if the stress levels are far below the elastic limit of the material due to the nucleation and progression of small cracks in some stressed sections of the file. The typical number of cycles to failure (NCF) is between some hundreds to several thousands [8]. This is equivalent to an expected life below some few minutes if a typical speed of rotation of 300rpm is considered.

There is no definitive conclusion about which is the predominant mechanism of failure in the clinical practice [6]. Satappan et al. [9] indicated a higher prevalence of torsional fracture (55.7%) than flexural fatigue (44.3%). However, Peng et al. [10] and Wei et al. [11] observed the opposite, with a clear preponderance of flexural fatigue. Notwithstanding, flexural fatigue seems to be the main concern for clinicians, because there is no easy method to avoid or anticipate this failure [7], resulting in a common practice of discarding the files after a certain number of uses to prevent it. However, there is no clear rule about the recommended number of uses, mainly due to the variety of factors potentially affecting NCF, such as root canal anatomy, file geometry or the operator’s experience, among others [12]. Therefore, a better understanding of the independent and combined effect of the different parameters on the flexural fatigue failure mechanism is desirable and additional research should be addressed to this end.

Experimental and simulated approaches have been used in the literature to analyse the effect of clinical and design parameters on the expected life of NiTi rotary files. Experimental approach has been mainly tackled by using in vitro studies in order to improve reproducibility. In general, those studies make the file rotate inside a curved path, reproducing the root canal geometry and registering NCF [6]. However, the differences among previous studies in the methodology and the setup used to bend the file hamper the comparability of results and limit their clinical relevance [6,12]. Due to this, a call for an international standard on the cyclic fatigue testing of rotary endodontic instruments is recurrent in the literature [6,13]. Despite these difficulties, the results from previous experimental studies on experimental fatigue tests on NiTi wires, or directly on endodontic files, have allowed drawing some conclusions about the fatigue behaviour of NiTi:The strain–life relationship is similar to that observed in low-cycle fatigue for metals, with a decrease in NCF for higher strain amplitudes, corresponding to highly curved canals [14].The fatigue life increases for files with a higher fraction of martensite, both by initial composition of the material or induced by phase transformation under deformation [15,16].The oral temperature and other parameters affecting the file temperature, such as rotational speed, can change the expected life, because they influence the phase fractions present in different points of the file in clinical use [17,18,19].Apart from the ‘structural fatigue’ resulting in the final fracture, NiTi exhibits ‘functional fatigue’, a significant and asymptotic change in the stress–strain curve and the phase transformation stresses during the first 100–140 cycles, resulting in a reduction in hysteresis cycle area and an increase in residual permanent strains after cycling [20].

These previous experimental studies have shown that, under constant value for other parameters, strain amplitude and NCF for NiTi wires are correlated, and this correlation can be adequately represented by the Coffin–Manson relation [14,16,20].

The simulated approach for analysing flexural fatigue has been mainly undertaken through the use of finite element (FE) models. FE analysis is a mathematical technique that can be used for predicting the state of stress and strain in a body or group of bodies under applied external loads and constraints. It is based on a fine discretisation of the geometry of bodies in a high number of small finite elements. This method allows gaining some insight into the stress and strain distributions inside the file, helping gain a better understanding of the failure mechanism. A recent study made a critical review of the use of this method applied to NiTi endodontic instruments [21] and highlighted some of the main limitations of the analyses performed to date. According to this study, very few studies modelled cyclic fatigue using FE simulation. The authors cited those of Lee et al. [8], Scattina et al. [7] and Ha et al. [22].

In [8] the authors performed a simulation of four different file models on three root canal geometries with different curvature and compared the results with those obtained from in vitro tests on equivalent systems. In the FE model, the file was rotated inside the simulated FE model of the root canal, and the maximum von Mises stress on the file nodes was analysed. They found that the location of the maximum von Mises stress in the FE model is a good predictor of the fractured section observed experimentally. Additionally, they confirmed a negative correlation between the maximum von Mises stress in the file and the NCF. The authors cited computational problems that forced them to reduce the rotational speed to 240rpm and to consider a friction coefficient of 0.01 in order to avoid nodal binding. The non-linear behaviour of the material was considered by using data from [23], which did not include the lower plateau in the stress–strain curve characteristic of the phase transformation for the unloading path, which corresponds to a lower stress level than that observed for the loading phase.

Scattina et al. [7] tried to predict NCF using FE simulations. They compared in vitro tests and FE simulations for three file models on a single root canal geometry. The model considered the contact between the file and the root canal, represented with rigid shell elements, and the simulation included the rotation of the file and the analysis of the stress state every 0.2 s during 2 s at a rotation speed of 300rpm. The authors used a multiaxial random fatigue criterion [24] to predict the NCF based on the stress history. They tuned the material properties with an optimisation procedure to match NCF predictions with experimental results on two of the file models and used these properties to predict NCF for the third model, finding a good agreement with experiments in both NCF and fracture location. However, the paper neither cited the final material parameters obtained from this optimisation nor the specific parameters considered.

In [22], the authors used FE simulation to develop a new file model intermediate between G-1 and G-2 models (Dentsply Maillefer, Ballaigues, Switzerland), but in this case the FE model did not include a fatigue simulation.

Cheung et al. [25] also performed an FE based fatigue analysis for comparing two different cross section geometries for the file, NiTi and steel based on a fully reversed bending analysis without including the root canal in the model. They applied the Coffin–Manson equation for predicting NCF.

The objective of the present study is to contribute to a better understanding of the effect of root canal geometry on the expected life of NiTi rotary files using FE simulation. To our knowledge, only Lee et al. [8] attempted a similar study, but they only considered three canal geometries with a different curvature, without changing the length of the straight part at the entrance of the root canal. Moreover, they based their analysis on the von Mises stress instead of analysing strain, which is the relevant parameter for predicting the fatigue life for low-cycle fatigue, according to the Coffin–Manson relation [25]. They also used a constitutive material model that did not include the hysteresis cycle formed in the stress–strain cycle due to the different stress levels corresponding to the phase transformation during loading and unloading.

In the present study, we used transient FE simulation for analysing the fatigue behaviour of a NiTi endodontic file with two different pitch values on a greater variety of root canal geometries, with changes in both the angle between the initial part and the apical part of the root and the radius of curvature in the connection between both sections. The model also includes a more comprehensive constitutive model for NiTi material and a very detailed discretisation of the file into quadratic finite elements. It simulates the introduction of the file into the canal and its rotation, including contact and friction. With this model, we calculated the strain range during a cycle for each point of the file. We used the Coffin–Manson relation to predict the expected NCF of the file in each root canal geometry.

## 2. Materials and Methods

The present investigation was conducted by using finite element analysis of a set of cases of study in which several combinations of endodontic rotary files and root canal geometries were studied.

Two different geometries of endodontic rotary file were considered, which are denoted as P2 (Figure 1a) and P3 (Figure 1b). Both of them have a convex ProTaper cross section shown in Figure 1c, their total length being $L_{total}=25\;\mathrm{mm}$, the length of their active part is $L_{ap}=16\;\mathrm{mm}$ and the diameter of their shaft and their tip is $d_{sh}=1.20\;\mathrm{mm}$ and $d_{ap}=0.25\;\mathrm{mm}$, respectively. The only difference between P2 and P3 resides in their axial pitch: $p_{z}=2\;\mathrm{mm}$ for P2 and $p_{z}=3\;\mathrm{mm}$ for P3.

On the other hand, the geometry of the root canal was constructed as follows (Figure 2):Segment AB has a length of $AB_{RC}=10\;\mathrm{mm}$, and it is perpendicular to the external surface, as indicated in Figure 2a. Line $L_{1}$ passes through point B, and it is inclined at angle $\theta_{RC}$ with respect to segment AB.A fillet, for which its radius is given by $r_{RC}$, is defined between segment AB and line $L_{1}$, as illustrated in Figure 2b. The tangency points of the fillet with the existing segments are denoted by D and E.Point F is located over line $L_{1}$ in such a manner that the total length from point A to point F is $L_{RC}=16\;\mathrm{mm}$. By performing this, the entire active part of the endodontic rotary files can be inserted within the canal. The resulting curve ADEF is the neutral axis of the root canal.Finally, a conic surface is created by sweeping a circumference along the neutral axis of the root canal, as illustrated in Figure 2c. At the entrance of the canal, the diameter of this circumference is $D_{RC}=1.26\;\mathrm{mm}$, and at the end of the canal it is $d_{RC}=0.26\;\mathrm{mm}$.

The resulting geometry of the root canal depends on two parameters: the angle of curvature $\theta_{RC}$ and the radius of curvature $r_{RC}$, according to the method proposed by Pruett [26]. The variation of these parameters allows us to consider different geometries for the root canal. In this study, three different values for the angle of curvature $\theta_{RC}\;=\;\left[30^{\circ },45^{\circ },60^{\circ }\right]$ and three different radii of curvature $r_{RC}\;=\;\left[5\;\mathrm{mm},10\;\mathrm{mm},15\;\mathrm{mm}\right]$ were considered. Combining these variables, 9 different geometries of the root canal were obtained, which are shown in Figure 3 (denoted as RC1, RC2, ..., RC9).

### 2.1. Definition of the Finite Element Model

Figure 4 shows the finite element model used in this study, which consists of an endodontic rotary file and a root canal. The root canal is modelled as a rigid surface under the assumption that its deformations are so small compared to the deformations of the endodontic rotary file that they can be neglected. The root canal remains immovable during the analysis.

The geometry of the endodontic file is generated and then discretised into quadratic finite element tetrahedrons following the ideas provided in [27]. The average element size has been set to 0.1mm, which has proven to provide a good compromise between accuracy and computational cost. The resulting finite element model has 103,609 nodes and 68,367 elements.

The top surface of the endodontic rotary file is defined as a rigid surface (shaded in grey in Figure 4), and its movements are coupled to the movements of a reference node that is used to define the boundary conditions of the endodontic rotary file.

The superelastic behaviour of the NiTi alloy was modelled by using the material model developed by Auricchio [28], which is summarized in Figure 5. Here, $E_{A}$ and $E_{M}$ represent the Young’s modulus of austenite and martensite, respectively. The beginning and the end of the loading transformation phase are given by $\sigma_{L}^{S}$ and $\sigma_{L}^{E}$, respectively, whereas the beginning and the end of the unloading transformation phase are given by $\sigma_{U}^{S}$ and $\sigma_{U}^{E}$. Finally, $\varepsilon_{L}$ represents the uniaxial transformation strain, and $\sigma_{ME}^{E}$ indicates the end of the martensitic elastic regime. In this work, the material properties that characterise this material model were extracted from [29].

The mechanical interaction between the root canal and the endodontic rotary file was considered by using a node-to-surface contact. A penalty-based constraint enforcement method was selected in order to enhance the convergency of the numerical solution. The tangential behaviour of the contact was also taken into account in the finite element model, with a constant coefficient of friction μ=0.1 [30].

The finite element model was solved by using transient analysis, which was conducted by using a large displacement formulation, and performed in two sequential steps:Insertion step: In the first step, the endodontic rotary file is inserted into the root canal. This is performed by prescribing a displacement at its reference node, which takes place along the *y* axis and has a magnitude equal to the length of the active part of the endodontic rotary file ($L_{ap}$). The rest of the movements of the reference node (displacements in *x* and *y* directions and all the rotations) are restricted in this step.Rotation step: In the second step, after the active part of the endodontic rotary file is inserted in the root canal, the endodontic rotary file performs a complete revolution along its axis of rotation. This is performed by prescribing a $360^{\circ }$ rotation along the *y* axis, while the rest of the movements of the reference node are restricted (rotations along *x* and *z* axes and all the displacements). The rotated angle is denoted by φ.

### 2.2. Fatigue Life Estimation from the Results of the Finite Element Analysis

The objective of this study was to predict the fatigue life of NiTi rotary files as they are rotating inside the root canal. For such a purpose, the strain results obtained from the rotation step of the finite element analysis were used in combination with the Coffin–Manson relation. This relation is conveniently expressed by the following equation:(1)$$\frac{\Delta \varepsilon }{2}=\varepsilon_{F}^{\prime }\cdot N_{f}^{c}+\frac{\sigma_{F}^{\prime }}{E}\cdot N_{f}^{b}$$
where $N_{f}$ is equivalent to NCF, $\varepsilon_{F}^{\prime }$ is the fatigue ductility coefficient, $\sigma_{F}^{\prime }$ is the fatigue strength coefficient, *c* is the fatigue ductility exponent, *b* is the fatigue strength exponent, Δε is the total strain range and Δε/2 is the strain amplitude. The prime in the equation indicates that the properties correspond to the cyclic properties, i.e., those after the initial 100–140 cycles.

In this equation, the first addend of the right side corresponds to the plastic strain amplitude $\Delta \varepsilon_{p}/2$ and the second one to the elastic strain amplitude $\Delta \varepsilon_{e}/2$. Figure 6 shows a logarithmic plot of the Equation (Equation 1), showing the contribution of these two terms, with the parameters for NiTi used in the present study, taken from [25]. The exponents *b* and *c* in the equation are negative, because the number of cycles correlates negatively with the strain amplitude. For high strain amplitudes, the plastic strain is much higher than the elastic strain, and the number of cycles to failure is low (low-cycle fatigue, LCF); for very low strain amplitudes, the second term of the equation is dominant because there is no significant plastic strain, and the number of cycles to failure is high (high-cycle fatigue, HCF). The transition between LCF and HCF can be observed as a change in the slope of the curve, which is typically located close to $10^{3}$–$10^{4}$ cycles.

Since the Coffin–Manson relation is based on a uniaxial strain, a criterion to reduce the obtained multiaxial strain state to an equivalent uniaxial strain condition is required. The critical plane concept has been extensively used for such a purpose, with successful results both for high and low cycle fatigue [31]. In the critical plane approach, the assessment of the fatigue failure is carried out in the material plane where the amplitude of some stress/strain components (or a combination of them) exhibits a maximum [24]. In the discretised finite element model of the endodontic rotary file, each node *i* on the surface will have an associated critical plane $\Pi_{i}$ characterised by its normal direction $\vec{n_{i}}$.

In this study, the critical plane $\Pi_{i}$ was defined in such a manner that its normal direction $\vec{n_{i}}$ is parallel to the direction of the maximum principal strain produced in node *i*, when the amplitude of this maximum principal strain reaches its maximum value. The direction $\vec{n_{i}}$ could be determined by observing the maximum principal strain at each frame of the analysis. However, in order to speed up the calculations, in this study it will be assumed that this maximum principal strain is normal to the plane that contains the trajectory of the observed node, as illustrated in Figure 7. Hence, $\vec{n_{i}}$ will be normal to the plane of rotation of node *i*.

After the critical plane $\Pi_{i}$ is determined for node *i*, a bending strain value $\varepsilon_{i,j}$ can be then obtained for that node at each analysis frame *j* by transforming the strain tensor and selecting the strain component in the direction of $\vec{n_{i}}$. Finally, the total strain range $\Delta \varepsilon_{i}$ for node *i* is defined as follows.
(2)$$\Delta \varepsilon_{i}=\max_{j=1..n}\left(\varepsilon_{i,j}\right)-\min_{j=1..n}\left(\varepsilon_{i,j}\right)$$

This total strain range $\Delta \varepsilon_{i}$ can be used in the Coffin–Manson relation to assess the fatigue life associated to node *i*. The material parameters considered for the application of the Coffin–Manson relation were extracted from [25].

The fatigue life of the endodontic rotary file was defined by the minimum value of fatigue life considering all the nodes in the surface of the endodontic rotary file. The node where $\Delta \varepsilon_{i}$ reaches a maximum value is the critical node, and it is denoted as i=crit.

## 3. Results

Figure 8 shows the strain results for endodontic rotary files P2 and P3 when they are rotating inside the most curved root canal RC9. Figure 8a shows the maximum principal strain plot for the case of study P2/RC9 in the instant of the rotation step of the analysis where the bending strain at the critical node reaches its maximum value. Figure 8b shows the minimum principal strain plot, for the same case of study, in the instant of the rotation step of the analysis where the bending strain at the critical node reaches its minimum value. The figure also shows the location of the critical node in both instants of time, as well as the critical plane for such a node. Figure 8c,d show the equivalent results for case P3/RC9 in which the file has a different pitch. The highest strains, both in tension and compression, are located at the edge in the surface of the file.

The evolution of the bending strain and the maximum and minimum principal strains during an entire rotation of the file are shown in Figure 9 for P2/RC9 and P3/RC9, reflecting a similar pattern for both files, but with a slightly higher strain range for the file with pitch 3mm. The phase shift is due to a different orientation of the edges of the file in the critical plane. The points marked with a star correspond to the frames of maximum and minimum bending strain values, shown in Figure 8.

The strain ranges obtained for all the studied cases are summarised in Figure 10. In these plots, the horizontal axis indicates the angle of curvature of the root canal and the vertical axis indicates the radius of curvature. The black dots indicate the combinations of radius and angle of curvature that have been studied, and the isolines are interpolated from these results.

Figure 11 shows the expected life for each file as a function of the radius and the angle of curvature of the root canal, calculated as indicated in Section 2.2, also with interpolated isolines.

The results in Figure 10 and Figure 11 indicate that root canals with higher curvatures (higher angles of curvature or/and smaller radii of curvature) force the files to a higher strain range and reduce their fatigue life. The effect of the angle of curvature is less significant for smaller radii of curvature, as indicated by the lower inclination of the isolines in the bottom of the figures. The file with pitch 3mm (P3) showed higher strain and shorter fatigue life as compared with that of pitch 2mm (P2), but this effect is slight, especially for less curved canals.

Finally, Figure 12 summarizes the effect of the degree of insertion of the endodontic rotary file inside the root canal for the case of study P2/RC9. Here, the degree of insertion of the rotary file inside the root canal is expressed as the percentage of the active part that is inserted. Figure 12a shows the evolution of the bending strain at the critical node during a entire revolution of the file for different degrees of insertion of the file inside the root canal (70%, 85% and 100%).

Figure 12b shows the evolution of the bending strain range with the degree of insertion of the endodontic rotary file inside the root canal. Here, additional data points are considered for a better observation of the evolution of this magnitude. This figure shows that the bending strain range is reduced as the degree of insertion of the endodontic rotary file inside the root canal is decreased.

## 4. Discussion

This study analysed numerically the effect of the root canal geometry on the bending fatigue life of NiTi rotary instruments. Some previous studies have shown the predictive capacity of this simulated approach based on the use of FE models to predict the location of the file fracture [7,8]. However, a recent review highlighted some limitations introduced in previous FE studies in this field, especially in the representation of the boundary conditions, the accuracy of the mesh to represent the real file geometry or the lack of consideration of friction in the FE analysis [21]. Our investigation solved these main limitations, with a very accurate mesh of quadratic elements for the file, and undertook a transient non-linear simulation of the introduction of the file inside the root canal and its rotation inside the canal, including contact and friction simulation. We also used the Coffin–Manson relation to estimate the expected life for the file working on a representative set of root canal geometries, proposing an adequate methodology for translating FE results to clinically relevant variables, which can be useful for manufacturers of NiTi rotary instruments.

Our results confirm that more curved canals are prone to reduce fatigue life, which was also previously observed in several in vitro studies [5,8,14]. We also found that the radius of curvature of the canal has a higher effect than the angle between the shaft and the apical portion of the file, especially for the lower radii of curvature. Changing the radius of curvature from 15mm to 5mm multiplies the strain amplitude by a factor of close to three for curvature angles of $30^{\circ }$ and by a factor close to 2.5 for curvature angles of $60^{\circ }$. Due to the logarithmic relationship between strain and NCF, this effect is higher in the expected life, which is reduced by a factor close to ten for $30^{\circ }$ of curvature angle and by a factor six for $60^{\circ }$. This result is in agreement with the experimental study by Chi et al. [5], although their setup was different because the curved part of the file reached the apical end for all conditions.

In most of the experimental studies, NCF is represented against a theoretical bending strain value obtained from geometrical approximations of the radius of curvature of the file and its diameter at the failure section using Equation (Equation 3) [14,16,17]:(3)$$\varepsilon_{a}=\frac{d}{2R}$$
where *d* is the file diameter, and *R* is the approximate radius of curvature of the file. Thus, the higher effect of the radius of curvature is expected from a theoretical point of view, because the ideal strain amplitude in bending is defined by Equation (Equation 3), depending only on the diameter of the file and the radius of curvature. However, the angle between the entrance and the apical portion of the root canal and also the clearance between the file and the canal walls affect the actual deformation of the file, which can be different to that defined by the geometrical approximation indicated by Equation (Equation 3), as observed in Figure 8. This would explain the effect observed for the angle of curvature.

The difference in strain amplitude and fatigue life for the files with pitch 2mm and 3mm is limited, with a slightly higher fatigue life for P2 and more pronounced for less curved root canals. Ha et al. [30] also observed lower stresses for closer pitch when analysing the screw-in tendency of NiTi rotary files with a transient FE model.

In Figure 12, we also studied the effect that the degree of insertion of the rotary file has on bending strain range. We observed that the bending strain range at the critical node decreases with a reduction in the degree of insertion. According to the Coffin–Manson relation, this increases the fatigue life of the rotary file. This reduction in the bending strain range could be explained because the diameter *d* of the rotary file at the curved part of the root canal is smaller with partial insertion and, according to Equation (Equation 3), this results in a reduction in the bending strain. Additionally, the greater clearance between the rotary file and root canal implies that the effective curvature radius of the file is higher than that of the root canal. Again, according to Equation (Equation 3), this implies an additional contribution to the reduction in bending strain range.

In this study, we based the analysis of file behaviour on the strain values observed, as recommended for LCF with significant strain values and fatigue life below some thousands of cycles [25]. We also proposed a method for calculating the strain amplitude during the cyclic rotation of the file, based on the use of a critical plane defining the direction of the maximum and minimum principal strains in the critical points. The results shown in Figure 9 reveal that the method proposed here for defining the critical plane is correct, because when the bending strain reaches its maximum and minimum points, its values coincide with the maximum and minimum principal strains, respectively.

This study has some limitations. We only considered a geometry of the file section, with a constant taper and pitch through its length. We also limited the analysis to root canals for which its geometrical axis can be represented by a planar curve. Moreover, despite our work trying to be as representative as possible of the clinical situation of the file rotating inside the root canal, some possible improvements remain and are commented in the following paragraphs. They can be observed as challenges for future studies trying to improve the accuracy of the model.

The material model used in this study is non-linear and included the phase transformation plateaus in the stress–strain curve and the different Young’s modulus of martensite and austenite. However, heat dissipated due to the hysteresis loop in the loading–unloading curve, and heat generated by sliding friction between the file and the canal can also induce phase transformations from martensite to austenite due to the shape memory effect of the material, adding a complex thermal–mechanical coupling that is not considered here. On the other hand, we have made estimations of the fatigue life based on the Coffin–Manson equation, but the fatigue ductility coefficient and fatigue ductility exponent were taken from the literature. We have not considered the effect of the phase transformations between austenite and martensite on fatigue response and, hence, on these parameters. Previous studies have shown that higher strains did not necessarily imply less cycles to failure, because a higher fraction of martensite results in a better response under cyclic loads [16]. To our knowledge, there is no clear approach for the moment to include this effect in the FE simulations.

In this study, we considered a transient simulation, but this simulation does not include the dynamical effects, which are dependent on the rotational speed of the file. Moreover, our model included normal and shear contact between the file and root canal walls, but these walls are simplified as rigid elements. Considering the dentine properties for the walls would be necessary for estimating the risk of canal transportation or ledging.

We have proposed a method for dealing with the multiaxial strain state in order to predict fatigue life, but the fatigue phenomenon in shape memory alloys is still under investigation [32], and there is no clearly established fatigue criterion for analysing multiaxial fatigue in those materials [7,32].

In addition, since the first introduction of NiTi rotary instruments in the last decade of the twentieth century, several changes have been introduced to new families of files in terms of composition, manufacturing methods and thermomechanical treatments, which is not always publicly known, affecting greatly the percentage of martensite or austenite present in the file in clinical use and also the fatigue life of instruments [15,33].

## 5. Conclusions

This numerical study confirms that the geometry of the root canal affects the fatigue life of rotary NiTi instruments. More curved root geometries, either by a higher inclination of the apical part of the canal with respect to the initial part at the entrance or by a lower radius of curvature, result in higher strain amplitudes in the file surface and to lower fatigue life. The radius of curvature in the curves of the root canal has a greater effect than the angle of curvature. Changes in the radius of curvature from 15mm to 5mm in the root canal can reduce fatigue life by factors close to ten. A change in the file pitch between 2mm and 3mm does not have an important effect on the fatigue life for the root canal geometries analysed, although the higher pitch exhibited a slightly lower life for root canals with low curvature. The degree of insertion of the file inside the root canal significantly affects the strain range obtained in the critical point of the file, and the strain range is reduced when the file is partially inserted within the file.

## Figures and Tables

**Figure 1 jcm-10-05692-f001:**
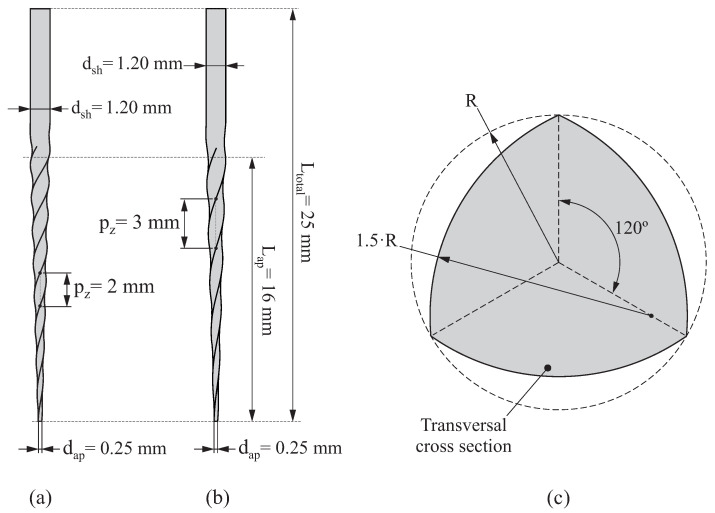
Geometry of the endodontic files P2 (**a**) and P3 (**b**) and normalised transversal cross section for both of them (**c**).

**Figure 2 jcm-10-05692-f002:**
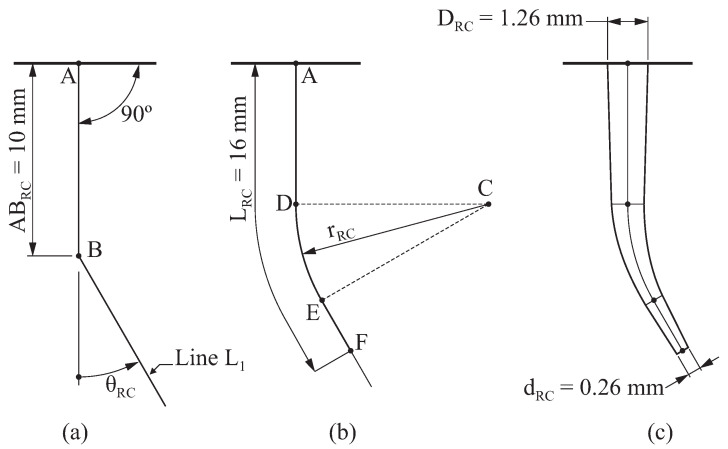
Parametrisation of the geometry of the root canal: (**a**) definition of the segment $AB_{RC}$ and line $L_{1}$, (**b**) definition of the fillet and (**c**) definition of the root canal surface.

**Figure 3 jcm-10-05692-f003:**
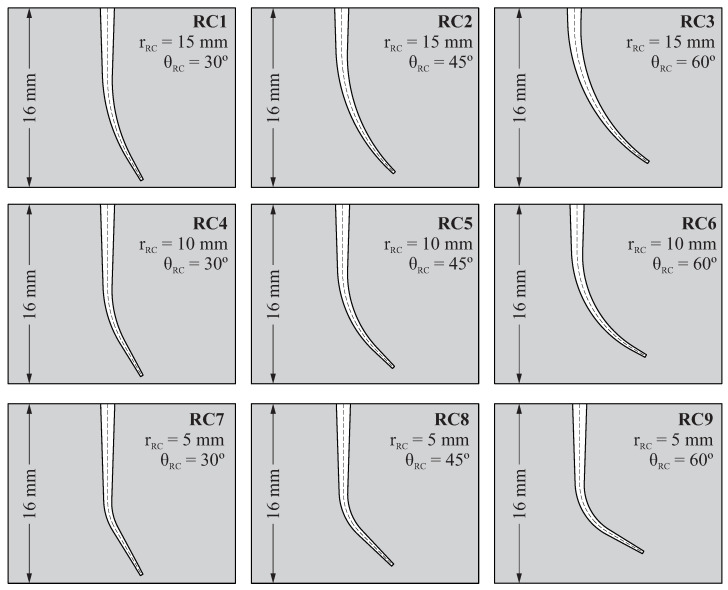
Geometries of root canal considered for the study.

**Figure 4 jcm-10-05692-f004:**
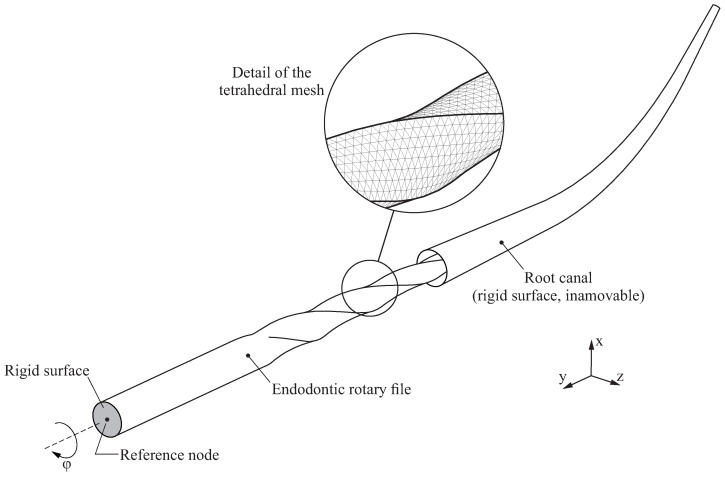
Definition of the finite element model.

**Figure 5 jcm-10-05692-f005:**
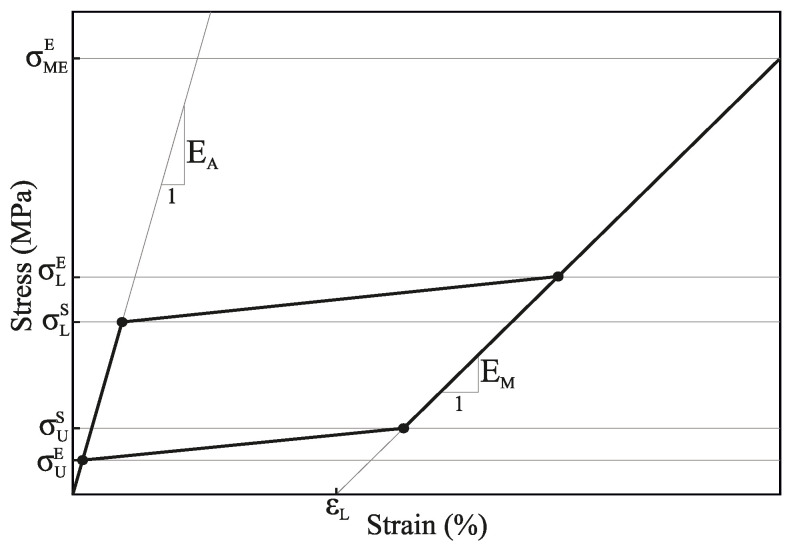
Definition of the stress–strain curve for the constitutive model of the superelastic NiTi alloy.

**Figure 6 jcm-10-05692-f006:**
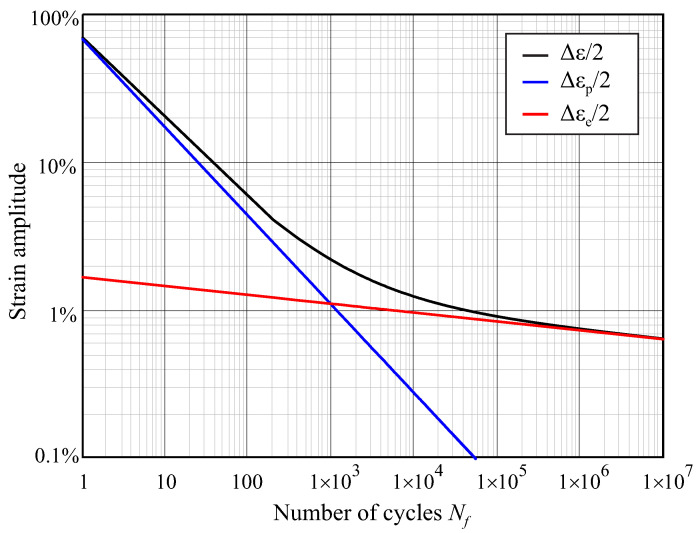
Coffin–Manson relation between strain amplitude and number of cycles to failure (NCF). Parameters for NiTi from [25]: $\varepsilon_{F}^{\prime }=0.68$, $\sigma_{F}^{\prime }=705\;\mathrm{MPa}$, E=42.5GPa, c=−0.6, b=−0.06.

**Figure 7 jcm-10-05692-f007:**
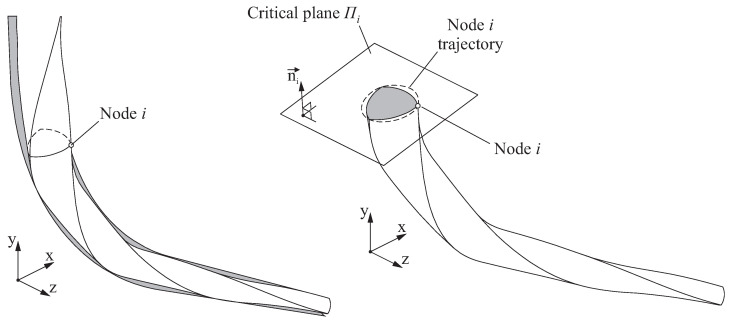
Determination of the critical plane and bending strain for node *i*.

**Figure 8 jcm-10-05692-f008:**
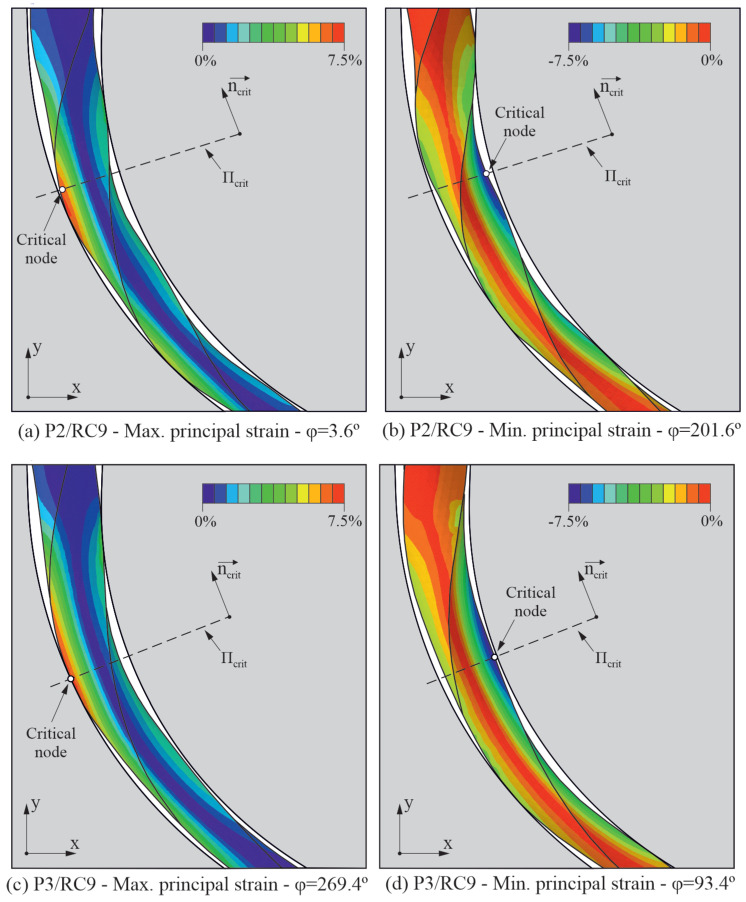
Principal strains in two different analysis frames for cases of study P2/RC9 and P3/RC9.

**Figure 9 jcm-10-05692-f009:**
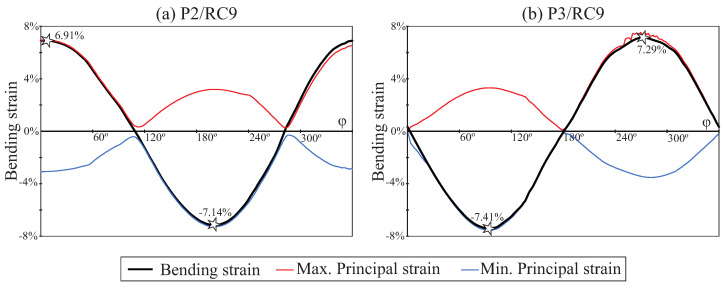
Strain history at critical nodes of cases of study P2/RC9 (**a**) and P3/RC9 (**b**).

**Figure 10 jcm-10-05692-f010:**
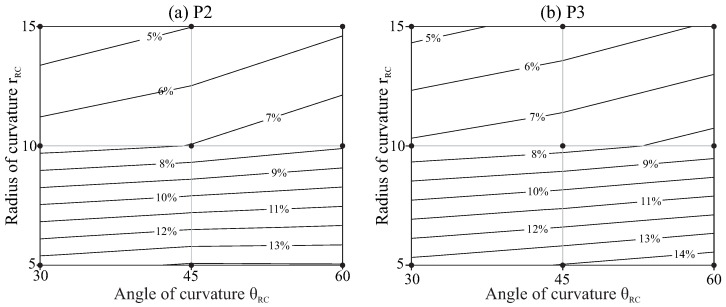
Maximum bending strain range as a function of the geometry of the root canal. (**a**) P2 and (**b**) P3.

**Figure 11 jcm-10-05692-f011:**
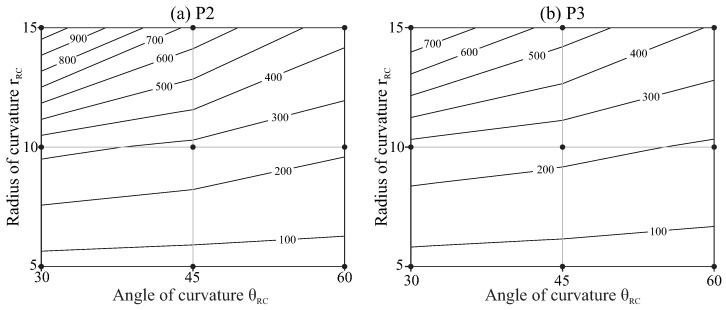
Expected life in number of fatigue cycles $N_{f}$ as a function of the geometry of the root canal. (**a**) P2 and (**b**) P3.

**Figure 12 jcm-10-05692-f012:**
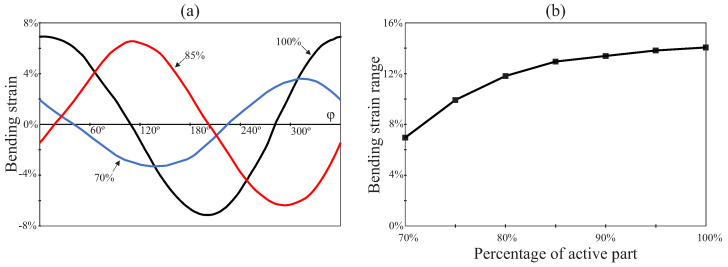
Effect of the degree of insertion of the endodontic file within the root canal. (**a**) Bending strain history at critical nodes and (**b**) bending strain range as a function of the degree of insertion.

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
