# Peer review of "Fatigue Analysis of NiTi Rotary Endodontic Files through Finite Element Simulation: Effect of Root Canal Geometry on Fatigue Life"

_jcm, 2021, doi:10.3390/jcm10235692_

Round 1
Reviewer 1 Report
This study used finite element simulation to analyze fatigue resistance of NiTi rotary files.
Questions:
How much up-and-down longitudinal motion can affect the mechanical wave proved by NiTI rotary files inside a canal?
Do the authors have related SEM evidence showing the pattern of fatigue failure (fatigue striations, dimple area etc.)?
Why not using double curvature model to analyze the fatigue resistance?
On page 10, line 295, the sentence was not completed. “…and reflected in Tab.??”. Please correct.
What is the limitation of the finite element simulation study?
Author Response
We would like to express our deep gratitude to the reviewer for his/her valuable comments.
To ease the work of the reviewers, we have highlighted in blue color all the modifications
done in the manuscript.
Question R1.1: How much up-and-down longitudinal motion can affect the mechanical wave
proved by NiTI rotary files inside a canal?
Answer A1.1: To answer this question, we have performed some additional analyses and
included new results in the manuscript (please see section 3, figure 12). In these analyses,
we have studied the influence of the degree of insertion of the rotary file inside the root canal
over the bending strain range. It is observed that, as the degree of insertion of the rotary file
is decreased, the bending strain range at the critical node is reduced and, in consequence,
the fatigue life of the rotary file is increased, according to the Coffin-Manson relation. Other
scenarios obtained as a result of combinations of different degrees of insertion could be
considered using cumulative fatigue criteria (i.e. Palmgren-Miner criteria). By including these
new results, we consider our work is more complete, and we thank the reviewer for this
question.
Question R1.2: Do the authors have related SEM evidence showing the pattern of fatigue
failure (fatigue striations, dimple area etc.)?
Answer A1.2: The objective of this work was to develop a method that allowed us to
understand the effect of the geometry of the root canal over the fatigue behaviour of the
endodontic rotary files through finite element simulation. In further works, we will expand our
work towards the investigation of the correlation between the simulation and the
experimental results.
Question R1.3: Why not using double curvature model to analyze the fatigue resistance?
Answer A1.3: At this stage of our investigation, we were interested in studying root canal
geometries that, being representative of the actual geometry of root canals, could be
parametrized using a small set of parameters. This has allowed us to consider a variety of
root canal anatomies by varying only two parameters, and to investigate the effect of these
two parameters on the fatigue life of the endodontic rotary files.
In future works, our interest is to continue this path of research by investigating the effect of
other root canal geometries on the mechanical response of the endodontic rotary files. In this
sense, we do agree with the reviewer that the consideration of root canals with double
curvatures could be an interesting topic to continue with our work, and we appreciate his/her
suggestion.
Question R1.4: On page 10, line 295, the sentence was not completed. “…and reflected in
Tab.??”. Please correct.
Answer A1.3: The reviewer is correct: there was a missing reference to Tab. 2. This mistake
has already been corrected in the revised version of the manuscript.
Question R1.5: What is the limitation of the finite element simulation study?
Answer A1.4: The finite element method is a trusted method to perform strength
calculations in many engineering fields. However, this method has some limitations that are
inherent to the method itself (for example, the typical balance between mesh refinement and
computational cost, or the convergence issues related to the non-linearities of the problem),
and these limitations are well known by most finite element analysts.
Other limitations are those related with the specific problem that we are solving using this
method. The most relevant of these limitations have already been discussed in the
manuscript (see last paragraphs of the discussion section). Other limitations are related to
the accuracy in which the parameters that are used to configure the method (coefficient of
friction, superelasticity parameters, etc.) are estimated.
All in all, finite element simulations allow us to gain insight into the behavior of the
endodontic file rotating within the root canal, and to predict stress and strain in any point of
the file, and hence its mechanical strength, with certain confidence margin. Moreover, in this
work we have managed to overcome some of the limitations that are typically encountered in
previous works, as explained in the manuscript.
Reviewer 2 Report
- the introduction is too long (like in a literature review) and more like a discussion in some parts
2. normally figures and tables must be limited to method and results sections only
3. in the discussion remove the bullets
Author Response
We would like to express our deep gratitude to the reviewer for his/her valuable comments.
To ease the work of the reviewers, we have highlighted in blue color all the modifications
done in the manuscript.
Question R2.1: The introduction is too long (like in a literature review) and more like a
discussion in some parts
Answer A2.1: Our purpose with this long introduction was to help readers approaching the
paper from different backgrounds and interests, such as clinicians, material scientists or
engineers. However, we agree with the reviewer that it can be reduced or reorganized
without losing important information for the paper. According to this, we have rearranged it,
suppressing the second paragraph, which provided tangencial information, and we also
moved the figures to the Materials and Methods section. We also moved to the Discussion
section the comments about original Eq. 2 (now Eq. 3), referred to how the strain is
estimated in most experimental studies. Finally, we also eliminated the division of the
introduction in several sections, because it is now possible to follow the reasoning without
them.
Question R2.2: Normally figures and tables must be limited to method and results sections
only
Answer A2.2: Following this and previous recommendations, we have moved the figures of
the Introduction to the Materials and Methods section. We think that this new structure is
clearer and improves the readability of the paper. Thanks for the suggestion.
Question R2.3: In the discussion remove the bullets
Answer A2.3: Following the reviewer’s recommendation we have removed the bullets and
rearranged the information of this part of the Discussion.